# Sex Differences in Exercise-Induced Bronchoconstriction in Athletes: A Systematic Review and Meta-Analysis

**DOI:** 10.3390/ijerph17197270

**Published:** 2020-10-05

**Authors:** Daniel Enrique Rodriguez Bauza, Patricia Silveyra

**Affiliations:** 1Clinical Simulation Center, The Pennsylvania State University College of Medicine, Hershey, PA 17033, USA; drodriguezbauza@pennstatehealth.psu.edu; 2Biobehavioral Laboratory, The University of North Carolina at Chapel Hill, Chapel Hill, NC 27713, USA

**Keywords:** inflammation, atopy, exercise-induced asthma, exercise-induced bronchoconstriction, sex differences

## Abstract

Exercise-induced bronchoconstriction (EIB) is a common complication of athletes and individuals who exercise regularly. It is estimated that about 90% of patients with underlying asthma (a sexually dimorphic disease) experience EIB; however, sex differences in EIB have not been studied extensively. With the goal of better understanding the prevalence of EIB in males and females, and because atopy has been reported to occur at higher rates in athletes, in this study, we investigated sex differences in EIB and atopy in athletes. A systematic literature review identified 60 studies evaluating EIB and/or atopy in post-pubertal adult athletes (*n* = 7501). Collectively, these studies reported: (1) a 23% prevalence of EIB in athletes; (2) a higher prevalence of atopy in male vs. female athletes; (3) a higher prevalence of atopy in athletes with EIB; (4) a significantly higher rate of atopic EIB in male vs. female athletes. Our analysis indicates that the physiological changes that occur during exercise may differentially affect male and female athletes, and suggest an interaction between male sex, exercise, and atopic status in the course of EIB. Understanding these sex differences is important to provide personalized management plans to athletes with underlying asthma and/or atopy.

## 1. Introduction

Asthma is one of the most common chronic non-communicable diseases of the airways, affecting about 339 million people worldwide [1]. The global prevalence of self-reported and physician-diagnosed asthma in adults is 4.3% (95% CI 4.2–4.4), with wide variation among countries [2]. Asthma is generally characterized by airway smooth muscle constriction (bronchospasm), excessive inflammation of the airway, and increased mucus production, although it presents in a variety of phenotypes and endotypes, ranging from mild and intermittent to severe and uncontrolled [3]. The diagnosis of asthma is determined by the history of respiratory symptoms, such as wheezing, shortness of breath, chest tightness, and cough, that vary over time and in intensity, together with variable expiratory airflow limitations [4,5].

Asthma is a heterogeneous disease, usually characterized by chronic inflammation. The clinical course of asthma is influenced by several factors, including genetics [6], environmental and occupational exposures [7], sex and gender [8], and hormones [9]. Atopy is also frequently associated with asthma [10]. Different from an allergy (i.e., the exaggerated immune response to a foreign antigen regardless of mechanism), atopy is characterized by an exaggerated IgE-mediated response to an allergen. Worldwide, 80% of childhood asthma and over 50% of adult asthma has been reported to be atopic [11]. In the United States, 56.3% of asthma cases have been attributed to atopy, a percentage that is greater among male patients than female patients [12]. 

Exercise and physical exertion are some of the most common triggers of bronchospasm in patients with asthma [13,14]. Bronchial hyperreactivity, a basic feature of bronchial asthma, occurs more often in athletes than non-athletes, especially in swimmers and winter sports athletes [15,16]. Exercise-induced respiratory symptoms usually involve acute narrowing of the airways that occurs during or after exercise and include exercise-induced bronchoconstriction (EIB) [17]. 

EIB is defined as the acute onset of bronchoconstriction occurring during or immediately after exercise [18]. Although EIB has been estimated to occur in up to 90% of patients with underlying asthma, it also occurs in subjects with no prior history of asthma and no symptoms outside of exercise [19]. Similarly, there is a subset of patients who have only exercise-induced asthma, but not chronic daily asthma [18]. Overall, while the epidemiology of asthma has been widely reported and studied worldwide, the epidemiology of exercise-induced asthma and EIB has not been well described. 

Sex and gender differences in the incidence, prevalence, and severity of lung diseases have been noted for years [20]. The terms “sex” and “gender” are oftentimes used interchangeably in research studies, although they represent different concepts. “Sex” refers to the underlying biological differences between males and females, including sex organs, XY chromosomes, and expression of endogenous hormones, while “gender” is a social construct that imparts roles and behaviors as masculine or feminine within the framework of historical or cultural contexts. The recent implementation of regulations and policies encouraging the incorporation of sex as a biological variable in research studies has permitted the identification and characterization of sex-specific mechanisms of lung diseases, including asthma, across the lifespan. Among children, the prevalence of asthma is higher in males than in females [21]. However, after puberty, the prevalence is about 20% higher in females than males, indicating a potential contribution of sex hormones [22]. Regarding the response to exercise, research studies evaluating sex differences have reported that the male and female body differ in cardiovascular, respiratory, thermoregulatory, metabolic, and neuromuscular responses that have clear implications for understanding sex-specific adaptations to exercise for athletic performance and overall health [23]. Gender, on the other hand, can potentially influence an individual’s behavior or preference towards a specific sport or physical depending on societal beliefs.

Although asthma has been widely reported to be more prevalent and severe in adult females than males [21,22], very few studies to date have addressed sex differences in EIB and/or the overall effects of exercise in male and female patients with asthma. One study reported that female adolescents, but not males, with EIB experience a lower health-related quality of life and poorer lung function than those without EIB [24]. Others reported that female athletes exhibit more severe symptoms of EIB than males, especially in the luteal phase of the menstrual cycle [25]. However, there is still no consensus in the literature on an established sexual dimorphism for EIB, nor studies addressing the mechanisms underlying potential sex differences in EIB prevalence and/or severity. Because minute ventilation rises with exercise [14], EIB likely results from changes in airway physiology triggered by the large volume of relatively cool, dry air inhaled during vigorous activity [26]. This is supported by research findings concluding that the main determinant of the occurrence and degree of bronchoconstriction is not the type of exercise, but rather the ventilation demand and humidity of the inspired air during exercise [27,28,29]. Here, we have conceptualized EIB as the acute onset of bronchoconstriction occurring during or immediately after exercise, independently of a subject’s underlying asthma. This concept is an accurate reflection of the disease’s underlying pathophysiology. The purpose of this review was to evaluate the overall prevalence of EIB and atopy in male and female athletes. We examined the available literature on sex differences in EIB in athletes and recreationally active individuals, outlining epidemiological data and results from clinical studies. We identified studies conducted in adult athletes and determined the prevalence of EIB, as well as the relationship between asthma, EIB, and atopy in male and female athletes.

## 2. Materials and Methods 

### 2.1. Literature Search

#### 2.1.1. Databases and Key Terms Searched

The literature search was guided by the Preferred Reporting Items for Systematic Reviews and Meta-Analyses (PRISMA) [30]. Studies were identified by searching PubMed/MEDLINE Complete, PubMed Central, and Google Scholar, up to July 2020. Search terms, phrases, and Medical Subject Headings (MeSH) were selected based on the purpose of the review and inclusion criteria. We used the search parameters: (‘exercise’ OR ‘athlete’) AND (‘gender’ OR ‘sex’) AND (‘asthma’ OR ‘bronchoconstriction’) AND (‘atopy’).

#### 2.1.2. Inclusion Criteria

The literature search was limited to human studies that were research-data based and published in only English language. Studies were selected if research study subjects were 12 years old and older, and athletes or recreational athletes were training for ≥ 2 days/week or 4 h/week in aerobic activities. Selected subjects’ training conditions were water sports (e.g., swimming, water polo), winter sports (e.g., cross-country skiing, biathlon, skeleton, alpine skiing, and ski cross), and other sports (including, but not limited to, running, speed skating, curling, handball, judo, triathlon, football, cycling, beach volley, rowing, athletics, sailing, badminton, canoeing, curling, equestrian, taekwondo, auto-racing, billiards, paragliding, rugby, tennis, roller hockey, kickboxing, fencing, basketball, or golf).

#### 2.1.3. Search Process and Study Selection

The literature search was conducted using both authors’ university library websites by entering search terms in the databases. Records were de-duplicated using the built-in mechanisms of the university library services and further completed manually. Articles were then assessed by their titles and abstracts for inclusion. Final selections were determined after full reading of articles. Each article was appraised based on the following five criteria: (1) relevance of the sampling strategy to address the research question, (2) representation of the sample on the target population, (3) appropriateness of the measurements, (4) risk of nonresponse bias, and (5) suitability of statistical analysis to answer the research question.

### 2.2. Data Extraction and Analysis

The following information was extracted from the studies: the first author of the study, the year of publication, the country of the study conducted, the purpose of the study, data on the sample size, details of the intervention, study quality, and measured outcomes. Main outcomes included EIB prevalence through questionnaire and/or pulmonary function testing (PFT). In addition, atopic status and self-reporting exercise-induced asthma-like symptoms in female (F) and male (M) athletes were extracted in all studies where these variables were reported. Atopy was defined as skin-test reactivity (skin prick test result). EIB was defined as a decrease of at least 15% in forced expiratory volume in one second (FEV_1_) vs. baseline at different timepoints after exercise spirometry or eucapnic voluntary hyperpnea (EVH), in addition to mannitol bronchoprovocation test or methacholine challenge; or having a prior medical diagnosis of EIB, respectively.

## 3. Results

A flow chart of the literature search is shown in Figure 1. The search string returned 1456 potentially relevant article citations. After systematically reviewing all the abstracts, 776 irrelevant studies and 508 duplicate papers were removed. Two independent investigators screened the remaining 172 full-text studies for eligibility and found that 35% of articles (60 studies including 7501 subjects; age range 12–67 years) met the inclusion criteria. These were categorized according to whether they reported sex differences in measured outcomes, and/or included male and female participants in the research design (Figure 2).

### 3.1. Prevalence of EIB in Athletes

To quantify the general prevalence of asthma and EIB in athletes, we first categorized the studies by method used for EIB diagnosis. These included exercise challenge [31,32], EVH with dry air [33], and EVH in combination with a bronchoprovocation test with mannitol or methacholine. We found that the majority of studies (43%), collectively enrolling 1829 subjects, used an exercise challenge (Figure 3), while 20% of the reviewed studies (*n* = 829 subjects) used EVH with dry air alone, or in combination with a bronchoprovocation test. In addition, 9% percent of studies used a bronchoprovocation challenge (*n* = 474 subjects), and 9% (*n* = 921 subjects) used self-reporting data (questionnaires) only. The remaining 20% of studies used a combination of these methods (*n* = 1050 subjects) and are further categorized in Figure 3. Overall, we found that the prevalence of EIB was reported in a total of 35 of the 60 reviewed studies, using a combination of the methods listed above. The studies, main findings, and ratio of male and female enrolled subjects are summarized in Table 1 [34,35,36,37,38,39,40,41,42,43,44,45,46,47,48,49,50,51,52,53,54,55,56,57,58,59,60,61,62,63,64,65,66,67,68]. Collectively, these studies enrolled 5103 athletes and reported a diagnosis of EIB in 1153 subjects. This corresponds to a general EIB prevalence of 23% (1153/5103).

### 3.2. Sex Differences in EIB Prevalence

Of the 60 studies focusing on EIB, 41 studies incorporated both male and female subjects, and only 19 reported sex differences in measured outcomes (Figure 4). However, not all studies focusing on EIB reported its prevalence; only 15 studies that included both males and females contained EIB prevalence data [43,44,51,53,55,58,61,68,69,70,71,72,73,74,75]. These studies collectively enrolled 2077 athletes and were 65% observational studies, 21% randomized control trials, and 14% clinical trials (Figure 4). The combined male to female ratio (M:F) in these studies was 1137:940, and the prevalence of EIB in this population was 30.3%. Sex differences in EIB prevalence were outlined as 17% in males and 13% females. To examine the relationship between EIB and subjects’ sex, we performed a Chi-square test of independence (Table 2) and found no significant differences in the relationship of these variables [χ^2^ (1, *n* = 2077) = 0.7654, *p* = 0.3811656].

### 3.3. Sex Differences in Atopic Status

We identified six studies reporting sex differences in EIB that also reported atopy status (determined by skin prick) (Figure 4) [51,65,69,71,72,74]. Combined, these included 980 participants (485 males, 495 females), including EIB-diagnosed athletes and healthy controls. Positive atopic status was reported in 323 subjects (33% prevalence) with an M:F ratio of 184:139 (Table 3). A Chi-square test of independence determined that the prevalence of atopy was significantly higher in male athletes than in female athletes, χ^2^ (1, *n* = 980) =10.7727, *p* = 0.00103 (Table 4).

### 3.4. Association of EIB and Atopy in Athletes

To evaluate the relationship between EIB and atopy in athletes, we analyzed studies reporting EIB and atopy, regardless of sex. We found five studies including 506 subjects combined, enrolling 262 athletes with EIB and 244 healthy athletes [61,69,71,72,74]. The prevalence of atopy was presented in 53% (*n* = 139) of EIB patients and 41% (*n* = 100) of healthy controls. A Chi-square test of independence indicated that the relationship between atopy status and EIB was statistically significant in athletes, when the sex variable is not considered, χ^2^ (1, *n* = 506) = 7.4, *p* = 0.006578 (Table 5).

### 3.5. Sex Differences in Atopic Status in Athletes with EIB

To assess sex differences in atopy in athletes diagnosed with EIB, we found four studies (*n* = 379 subjects combined) where both sex and EIB diagnosis was reported [44,71,72,74]. We observed that positive atopic status was more than twice as high in male (36%) than female (15%) athletes with EIB. Conversely, no sex differences were observed when comparing non-atopic male and female athletes with EIB (Table 6). A Chi-square test of independence revealed that the relationship between sex and atopy in EIB athletes was statistically significant, χ^2^ (1, *n* = 379) =16.2439, *p* = 0.000056 (Table 7).

## 4. Discussion

Exercise-induced bronchoconstriction occurs in the presence or absence of clinically recognized asthma. A sex disparity exists in individuals suffering from asthma throughout life, calling for the question of whether sex also influences EIB. In this review, we studied the prevalence of EIB in adult professional and recreational athletes and its relationship with atopic status in males and females enrolled in 60 studies. Our analysis of the literature confirmed the previously described positive association of atopy and EIB in athletes [68] and showed that while the prevalence of EIB does not display an overall sex dimorphism, atopy is more prevalent in male athletes than in female athletes. Moreover, our analysis indicates that male athletes are twice as likely to present with atopic EIB than females. These results indicate that potential sex-specific mechanisms exist in the inflammatory and physiological changes triggered by exercise in athletes.

One of the major triggers for bronchoconstriction in a vulnerable subject is water loss during periods of high ventilation. Strenuous exercise creates a hyperosmolar environment by introducing dry air into the airway with compensatory water loss, leading to transient osmotic changes in the airway surface. This hyperosmolar environment leads to mast cell degranulation and eosinophil activation with consequent release of inflammatory mediators, including leukotrienes. This process triggers bronchoconstriction and inflammation of the airway, as well as stimulation of sensory nerves and release of neurokinin and mucins [75]. Prior studies in animal models and human cells have reported sex differences in mast cell functionality and suggested a potential regulation by sex hormones [76,77,78,79,80]. Mast cells expressing estrogen, progesterone, and androgen receptors have also been identified in the human upper airway and nasal polyps, indicating that this may be a major route for the involvement of sex hormones in exercise-induced airway inflammation [81,82]. Similarly, studies have suggested a relationship between mast cell-derived mediators, sex hormones, and the development of asthma and allergic lung disease [83]. The biosynthesis of leukotrienes and other pro-inflammatory eicosanoids and prostaglandins by mast cells is also sex-biased and has been shown to be mediated by androgens [84,85,86]. At the neural level, sex differences in neurokinins and their receptors have been reported in adults [87,88], and sex hormones have been shown to regulate neurokinin-dependent activation of airway smooth muscle in allergic asthma [89,90,91]. Together, these studies illustrate the complexity of mechanisms involved in EIB in males and females and suggest a potential regulation by sex steroids at the immune and neural level.

Compared to the general population, elite athletes have a higher prevalence of EIB that varies with the intensity of exercise and the environment [75]. Increased bronchial responsiveness and asthma are strongly associated with atopic disposition and its severity in elite athletes [92], and atopic diseases are overall more common in athletes [93]. Our analysis of the literature revealed that 23% of the athlete population studied presented with EIB. Interestingly, the severity and prevalence of EIB was even higher (30.3%) in studies that reported sex differences. Our data also concur with prior research reporting a greater prevalence of EIB in high-performance athletes than in the general population. These studies suggested that prolonged inhalation of cold, dry air, and airborne pollutants are some of the factors influencing the high prevalence of EIB in athletes [14]. Other studies have reported a prevalence of EIB between 30–70% among elite or Olympic-level athletes [19,58], as opposed to 5–20% in the general population [75,94]. While EIB is frequently documented with asthma and reflects insufficient control of underlying asthma, few epidemiological studies of EIB have categorized participants by asthma status. Thus, the true prevalence of EIB within the non-asthmatic general population has not been fully established, preventing researchers from evaluating and quantifying sex and/or gender differences as well. In this regard, a sex disparity in EIB and airway hyperresponsiveness (AHR) has been reported in young adults, with lower rates of mild AHR in males vs. females but higher rates of moderate AHR and atopy in males [95]. In the United States, 56.3% of asthma cases are attributable to atopy, a percentage that is greater among males than females [12,96]. Repeated and strong exposure to pollen and other allergens causes bronchial and upper respiratory symptoms in athletes, although very few studies have investigated occurrence of atopic status in athletes [97]. Our analysis revealed that the male sex and atopic status are potential risk factors for EIB in athletes. We also identified a 2:1 ratio of atopic EIB in male vs. female athletes. However, female athletes were overall underrepresented in studies assessing EIB, as is the case for studies in many lung diseases [98].

Estimating the prevalence of EIB has also been problematic due to the lack of a gold standard for diagnosis. Since 2016, a joint taskforce (JTF) including the American Academy of Allergy, Asthma and Immunology, the American College of Allergy, Asthma and Immunology, and the Joint Council of Allergy, Asthma and Immunology [97] has recommended that the diagnosis of EIB should rely on performing a standardized bronchoprovocation challenge (exercise or a surrogate), because the presentation of EIB will vary with the type of challenge and the conditions under which the challenge is performed. In our review of the literature, we found that 43% of the selected papers used an exercise challenge alone as a diagnostic tool, whereas 14.2% used it in combination with a bronchial provocation test, such as mannitol or methacholine. While we do not know the exact conditions in which these tests were performed, there is a possibility that a sex bias exists in their ability to serve as EIB diagnostic tools. In this regard, 9% of the studies included used self-reporting data via questionnaires. The JTF recommends that a diagnosis of EIB is confirmed by demonstration of airway reversibility or challenge in association with a history consistent with EIB, because self-reported symptoms are not always accurate [73,97]. 

Our study has several limitations. First, while we took measures to minimize bias, there is a risk of bias assessment associated with the protocols used to retrieve and select the literature, including selective reporting of findings in abstracts, and variations in the nomenclature and definitions used in studies [30]. Furthermore, the analysis conducted in this report showed an overall higher rate of self-reporting EIB asthma-like symptoms in females than males, indicating that the sex differences observed in atopic EIB in athletes may be even more striking due to underreporting of symptoms by male athletes. Moreover, none of the studies accounted for the menstrual phase of female participants, nor for oral contraceptive use at the time they were surveyed. This represents a limitation since hormonal status has been shown to potentially alter self-reported asthma like symptoms in women [98,99]. Another limitation of the current analysis is that the protocols for determining the presence and magnitude of EIB were not consistent among the studies included, as shown in Figure 3. While there are not reports comparing the accuracy of these methods in diagnosing EIB in males and females, it is possible that variations in EIB prevalence across studies are a result of variability in these measurements’ validity. Our analysis also incorporated data from studies enrolling both professional and recreational athletes who exercised at a specific frequency and intensity, according to our inclusion criteria. While sports medicine experts still have not reached a consensus on a defined nomenclature to describe training intensity, regularity, and/or competitiveness level in athletes and non-athletes [100], it is possible that combining all regular exercisers in our analysis led to missing potential differences in EIB/atopy prevalence in men and women who are professional athletes vs. recreational exercisers. 

Recent studies have yielded significant advances in our understanding of how intrinsic and extrinsic factors can impact airway function in athletes. Extrinsic factors include environmental exposure to temperature, humidity, aeroallergens, irritants, and pollution. Intrinsic factors include atopy, allergic rhinitis, asthma, body mass index, and airway anatomy. These factors can affect both the athlete’s quality of life and athletic performance, but also contribute to sex differences in exercise physiology and EIB [101]. In this regard, the menstrual cycle phase is an important determinant of EIB severity in female athletes with mild atopic asthma [102]. An estimated 33–52% of females with asthma report a premenstrual worsening of asthma symptoms, and an additional 22% report asthma that is worse during menses [102]. However, the temporal correlation between asthma symptoms and steroid levels does not provide a simple answer as to whether estrogen and/or progesterone improve or worsen asthma. Female sex steroid hormones could affect exercise capacity and performance through numerous psychologic mechanisms, including substrate metabolism, cardiorespiratory function, and thermoregulation [103,104,105,106]. Thus, hormone level changes may lead to either improved or decreased performance at various times throughout the menstrual cycle [107,108]. It is also possible that the reduction in estrogen levels and other menstrual cycle disturbances that occur with exercise are associated with the lower prevalence of atopic EIB observed in female athletes [26,109]. Overall, the relationship between exercise and menstrual cycle is an important variable to consider when analyzing sex differences in EIB and atopy. Future research studies in animal models and human subjects should assess the factors predisposing athletes to develop concurrent atopy and EIB, as well as the mechanisms underlying different outcomes in males vs. females. Overall, more research that accounts for sex as a biological variable and incorporates anatomical and/or physiological factors (e.g., lung volumes, hormone fluctuations) is needed to better understand sex-specific mechanisms of EIB, and potentially develop sex-specific therapeutics to prevent and treat EIB and other pulmonary conditions [110].

## 5. Conclusions

In conclusion, we show here that a relationship exists between the male sex and atopic status in the course of EIB in athletes. Understanding sex differences in EIB and atopy in athletes could lead to the development of better personalized training and disease management plans for athletes with these underlying conditions. 

## Figures and Tables

**Figure 1 ijerph-17-07270-f001:**
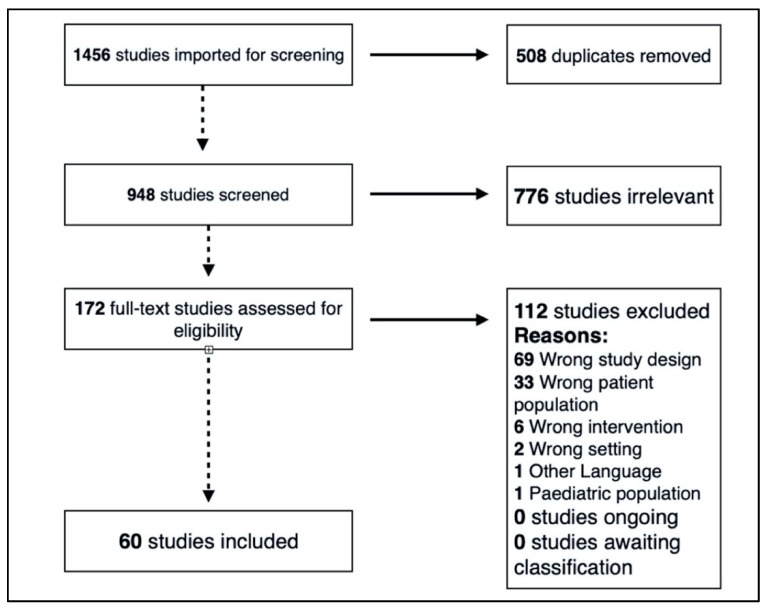
Preferred Reporting Items for Systematic Reviews and Meta-Analyses (PRISMA) flow diagram of literature search and selection process.

**Figure 2 ijerph-17-07270-f002:**
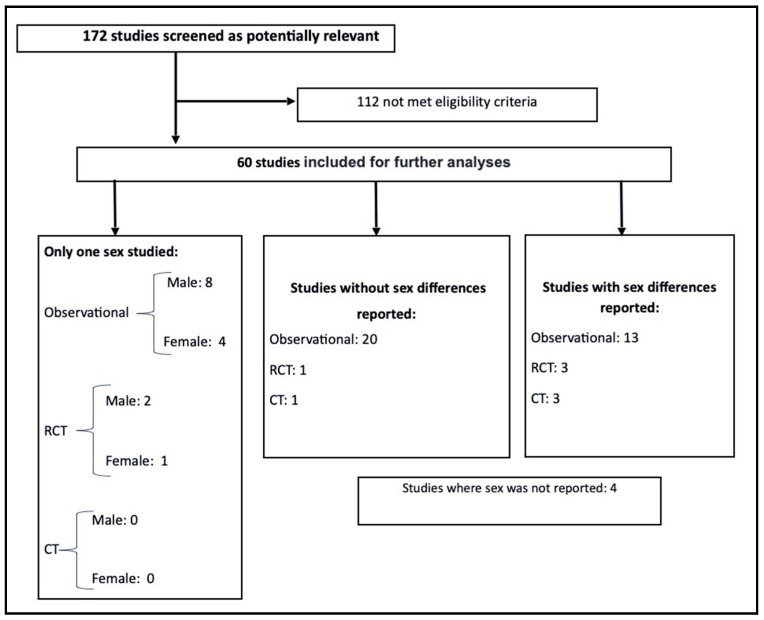
Screening process and classification of articles by sex reporting. RCT: randomized controlled trial; CT: clinical trial.

**Figure 3 ijerph-17-07270-f003:**
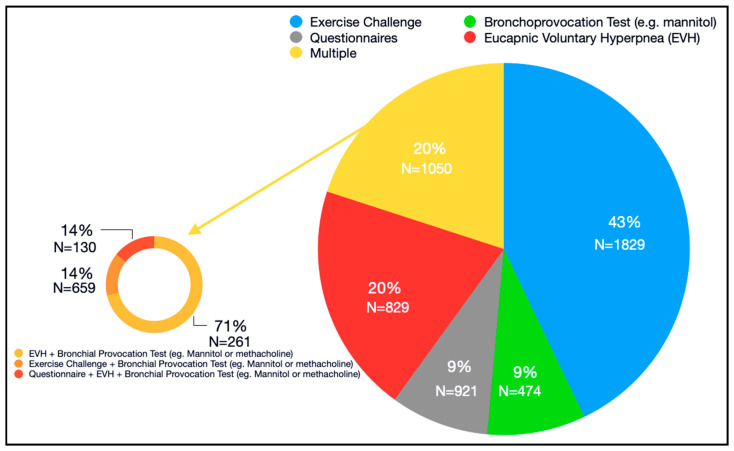
Diagnostic tests used to determine exercise induced bronchoconstriction in selected articles. EVH: eucapnic voluntary hyperpnea.

**Figure 4 ijerph-17-07270-f004:**
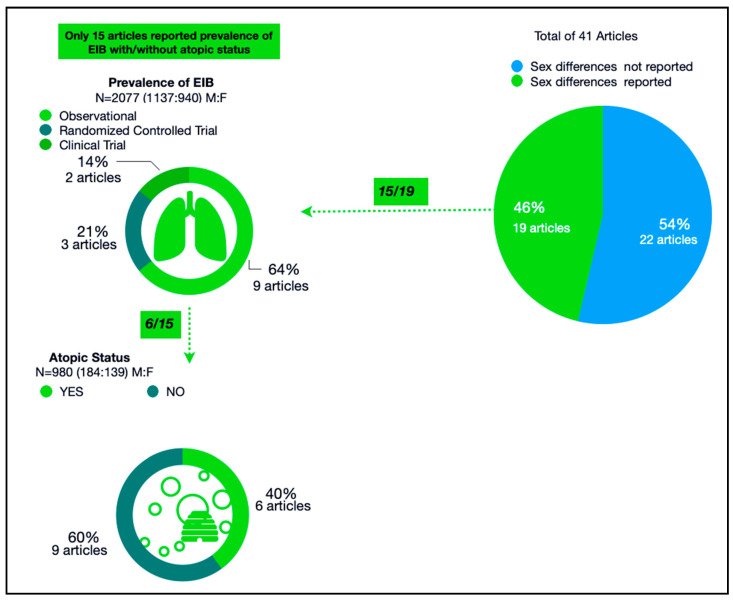
Articles reporting sex differences and atopic status in exercise-induced bronchoconstriction (EIB).

**Table 1 ijerph-17-07270-t001:** Prevalence of exercise-induced bronchoconstriction (EIB) among male and female athletes.

Author, Year	Method for EIB Diagnosis	Study Population (*n* = subjects), Sex Ratio	EIB Prevalence
Ahad, Sandila, and Siddiqui, 2004 [34]	Exercise challenge	Pakistani hockey players (*n* = 27) Male only	19%
Ahad, Sandila, Siddiqui, and Ahmed, 2003 [35]	Exercise challenge	Pakistani athletes (*n* = 179) Male only	7%
Allen et al., 2019 [36]	EVH ^1^	Recreational athletes (*n* = 180) (120:60) M:F	37%
Ansley, Kippelen, Dickinson, and Hull, 2012 [37]	EVH and Bronchoprovocation Test (dry powder mannitol)	UK soccer players (*n* = 65) Male only	51%
Becerril-Ángeles et al., 2017 [38]	Exercise challenge	Mexican high school and college athletes of summer sports (*n* = 208) (115:93) M:F	7.2%
Bonini et al., 2015 [39]	Exercise challenge and Bronchoprovocation Test (dry powder mannitol or methacholine)	Italian Olympic Delegation at Summer (Sydney 2000, Beijing 2008, and London 2012) and Winter (Vancouver 2010) Olympics (*n* = 659) (441:218) M:F	14.7%
Bougault, Turmel, and Boulet, 2010 [40]	EVH and Bronchoprovocation Test (dry powder methacholine)	Swimmers and winter sport athletes (*n* = 45 in each group) (39:51) M:F	75% in swimmers 40% in winter sport athletes
Burnett, Burns, Merritt, Wick, and Sharpe, 2016 [41]	Exercise challenge	80 college athletes (56:24) M:F	42.5%
Burnett, Vardiman, Deckert, Ward, and Sharpe, 2016 [42]	Questionnaire	196 college athletes (56:140) M:F	28.6%
Couillard et al., 2014 [43]	Questionnaire, EVH and Bronchoprovocation Test (dry powder methacholine)	130 athlete swimmers (*n* = 51 swimmers, *n* = 10 synchronized swimmers), winter athletes (*n* = 30 cross-country skiers, *n* = 11 speed skaters training outdoors, *n* = 9 biathletes), other endurance sports athletes (*n* = 10 triathletes, *n* = 7 cyclists, *n* = 2 canoe-kayakers) (65:65) M:F	51%
Couto et al., 2015 [44]	Bronchoprovocation Test (dry powder mannitol or methacholine)	Portuguese and Norwegian athletes training at high-competitive levels (national, international, or Olympic teams) (*n* = 324) (107:43) M:F	46.2%
Dickinson, McConnell, and Whyte, 2011 [45]	EVH	Elite British athletes (*n* = 228) Sex not reported	34%
Durand et al., 2005 [46]	Exercise challenge	Ski-mountaineering athletes (*n* = 31) (28:3) M:F	48.3 %
Hallstrand et al., 2002 [47]	Exercise challenge	Adolescents participating in organized sports from three suburban high schools (*n* = 256) (136:120) M:F	9.4%
Hunt et al., 2017 [48]	Exercise challenge	*N* = 92 players from three senior inter-county hurling teams Male only	9.8%
Kippelen, Caillaud, Coste, Godard, and Préfaut, 2004 [49]	Exercise challenge	*n* = 97 athletes Male only	5.3%
Kukafka et al., 1998 [50]	Exercise challenge	High school football players (*n* = 238) Male only	9%
Langdeau et al., 2009 [51]	Bronchoprovocation Test (dry powder methacholine)	*n* = 100 athletes (65:35) M:F	AHR higher in females (60%) vs. males (21.5%), *p* < 0.0001
Leuppi, Kuhn, Comminot, and Reinhart, 1998 [52]	Bronchoprovocation Test (dry powder methacholine)	Elite ice hockey players (*n* = 26) and floorball players (*n* = 24) Male only	34.6% (ice hockey); 20.8% (floorball players)
Levai et al., 2016 [53]	EVH	38 boxers (33:5) M:F 44 swimmers (25:19) M:F	68% (elite swimmers); 8% (boxers)
Lund, Pedersen, Larsson, and Backer, 2009 [54]	Questionnaire	329 elite athletes (198:131) M:F	55%
Molphy et al., 2014 [55]	EVH	Recreationally active individuals (*n* = 136)	13.2%
Norqvist, Eriksson, Söderström, Lindberg, and Stenfors, 2015 [56]	Questionnaire	*n* = 402 Swedish elite skiers, orienteers, and former Olympic athletes (cross-country and biathlon) (218:184) M:F	11%
Osthoff et al., 2013 [57]	EVH and Bronchoprovocation Test (dry powder mannitol)	*n* = 44 athletes aiming to participate at the 2008 Beijing Paralympic Games (30:14) M:F	20%
Parsons et al., 2012 [58]	EVH	*n* = 144 athletes from six different varsity sports at a large National Collegiate Athletic Association Division I collegiate athletic program (79:65) M:F	3%
Pedersen, Winther, Backer, Anderson, and Larsen, 2008 [59]	EVH and Bronchoprovocation Test (dry powder methacholine)	16 elite swimmers Female only	50%
Pohjantähti et al., 2005 [65]	Exercise challenge	*n* = 20 healthy elite cross-country skiers (14:6) M:F *n* = 18 non-asthmatic controls (7:11) M:F	45%
Rundell, Spiering, Evans, and Baumann, 2004 [60]	Exercise challenge	United States national ice hockey team players (*n* = 43) Female only	21%
Rundell et al., 2003 [61]	Exercise challenge	*n* = 18 elite athletes, cross-country skiers (13:5) M:F	50%
Sallaoui et al., 2007 [62]	Exercise challenge	*n* = 107 elite athletes (63:44) M:F	13%
Sallaoui et al., 2009 [63]	Exercise challenge	*n* = 326 athletes (188: 138) M:F	9.8%
Sallaoui et al., 2011 [64]	Exercise challenge	*n* = 107 elite athletes (63:44) M:F	13%
Seys et al., 2015 [66]	EVH	Swimmers (*n* = 26), indoor athletes (basketball/volleyball, *n* = 13), and controls (not exercising more than 4h/week, *n* = 15). Swimmers (16:10) M:F Indoor athletes (8:5) M:F Controls (7:8) M:F	23% of swimmers, 0% of indoor athletes, 1% of controls
Stenfors, 2010 [67]	EVH and Bronchoprovocation Test (dry powder mannitol or methacholine)	*n* = 46 Swedish elite cross-country skiers (24: 22) M:F	17%
Teixeira et al., 2012 [68]	EVH	20 Brazilian long-distance runners. Male only	25%

EVH: Eucapnic voluntary hyperpnea; AHR: airway hyperresponsiveness; M: male, F: female.

**Table 2 ijerph-17-07270-t002:** Prevalence of EIB in studies incorporating male and female athletes.

Sex/EIB	EIB	Healthy	Marginal Row Total
Male	350 (339.03) [0.36] ^1^	779 (789.97) [0.15]	1129
Female	268 (278.97) [0.43]	661 (650.03) [0.19]	929
Marginal Column Total	618	1440	2058 (Grand Total)

^1^ Numbers indicate: observed cell total (expected cell total) [cell Chi-square statistic].

**Table 3 ijerph-17-07270-t003:** Sex differences in atopic status.

Atopic Status/Sex	Male	Female	Row Total
Positive atopic status	184 (19%)	139 (14%)	323 (33%)
Negative atopic status	301 (30%)	356 (36%)	657 (67%)
Column Total	485	495	980 (Grand Total)

**Table 4 ijerph-17-07270-t004:** Chi-square statistic of athletes’ atopic status.

Atopic status/Sex	Male	Female	Row Totals
Positive atopic status	184 (159.85) [3.65] ^1^	139 (163.15) [3.57]	323
Negative Atopic status	301 (325.15) [1.79]	356 (331.85) [1.76]	657
Column Totals	485	495	980 (Grand Total)

^1^ Numbers indicate: observed cell total (expected cell total) [cell Chi-square statistic].

**Table 5 ijerph-17-07270-t005:** Atopic status in athletes with and without EIB.

Atopy/EIB	EIB	Healthy Controls	Total Row
Atopy	139 (123.75) [1.88] ^1^	100 (115.25) [2.02]	239
No atopy	123 (138.25) [1.68]	144 (128.75) [1.81]	267
Total column	262	244	506 (Grand Total)

^1^ Numbers indicate: observed cell total (expected cell total) [cell chi-square statistic].

**Table 6 ijerph-17-07270-t006:** Sex differences in atopic status in athletes with diagnosis of EIB.

EIB Athletes	Atopy	No Atopy	Total Row
Male	137 (36%)	92 (24%)	229 (60%)
Female	58 (15%)	92 (24%)	150 (40%)
Total Column	195	184	379 (Grand Total)

**Table 7 ijerph-17-07270-t007:** Relationship between atopy and sex in athletes with EIB.

	Atopy	No Atopy	Marginal Row Totals
Male	137 (117.82) [3.12] ^1^	92 (111.18) [3.31]	229
Female	58 (77.18) [4.77]	92 (72.82) [5.05]	150
Marginal Column Totals	195	184	379 (Grand Total)

^1^ Numbers indicate: observed cell total (expected cell total) [cell Chi-square statistic].

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
