# Peer review of "Sex Differences in Exercise-Induced Bronchoconstriction in Athletes: A Systematic Review and Meta-Analysis"

_ijerph, 2020, doi:10.3390/ijerph17197270_

Round 1

Reviewer 1 Report

Major comments

The Authors address issue important in practice not only of sports medicien doctors but also general physicians. Justification and design do not have considerable flaws and the introduction of the papaer thoroughly describes the problem and current knowledge of the EIB and asthma in athletes. Inclusion criteria for papers and literature search and selection proces are clearly defined and described in sufficient detail. Also, the approach and strategy of data analysis seems correct and justified.

This review finally included 60 papers referring to different groups of athletes at different levels of competitiveness. Recently, there has been much debate (still not concluded, though) regarding the nomenclature used for description of training intensity, regularity and competitiveness level. Therefore, some confudion exists in what regards terms „elite’, „high-performance”, „competitive” , „professional” used with regard to regular exercisers. Moreover, physical activity gains wider popularity and recognition among general public which implies increased numer of people of different ages and fitness levels taking on regular, sometimes strenuous, physical exercise.

These facts are additional aspects that render the issue of exercise-associated respiratory symptoms even more complicated. In the present review, this aspect seems somewhat omitted. However, the authors included also studies referring to recreational athletes in their analysis. In my opinion, potential differences of EIB prevalence in recreational exercisers should be mentioned in the discussion/commentary.

.

Minor issue

Section 3.2 lines 159-159

It is completely clear what is meant by the phrase „However, only 15 studies 158 that included both males and females reported EIB prevalence” whereas the authors stated that they intended to include studies focusing primarily on EIB. Does this phrase refer to separate reports on prevalence in males and females? Please clarify.

Otherwise, the review does not contain major flaws and clearly and thoroughly describes the problem with conlusions correctly derived from the presented facts. It is of importance for medical public and I recommend it to be published after suggested clarifications are addressed

Author Response

Reviewer 1 

Major comments

The Authors address an issue important in practice not only of sports medicine doctors but also general physicians. Justification and design do not have considerable flaws and the introduction of the paper thoroughly describes the problem and current knowledge of the EIB and asthma in athletes. Inclusion criteria for papers and literature search and selection process are clearly defined and described in sufficient detail. Also, the approach and strategy of data analysis seems correct and justified.

This review finally included 60 papers referring to different groups of athletes at different levels of competitiveness. Recently, there has been much debate (still not concluded, though) regarding the nomenclature used for description of training intensity, regularity and competitiveness level. Therefore, some confusion exists in what regards terms “elite'', “high-performance”, ”competitive” , “professional” are used with regard to regular exercisers. Moreover, physical activity gains wider popularity and recognition among the general public which implies an increased number of people of different ages and fitness levels taking on regular, sometimes strenuous, physical exercise.

These facts are additional aspects that render the issue of exercise-associated respiratory symptoms even more complicated. In the present review, this aspect seems somewhat omitted. However, the authors also included studies referring to recreational athletes in their analysis. In my opinion, potential differences of EIB prevalence in recreational exercisers should be mentioned in the discussion/commentary.

Response: We thank the reviewer for their critical assessment of our paper. We have modified the discussion to expand on the implications of our findings for professional vs. non professional exercisers (page 11, lines 299-305).

Minor issue

Section 3.2 lines 159-159

It is completely clear what is meant by the phrase „However, only 15 studies that included both males and females reported EIB prevalence” whereas the authors stated that they intended to include studies focusing primarily on EIB. Does this phrase refer to separate reports on prevalence in males and females? Please clarify.

Response: While we included 60 studies focusing on different aspects of EIB, only a fraction of these studies reported the prevalence of EIB (and sex differences) in the populations of study. We have edited this sentence to avoid confusion (page 8, lines 172-174).

Otherwise, the review does not contain major flaws and clearly and thoroughly describes the problem with conclusions correctly derived from the presented facts. It is of importance for medical public and I recommend it to be published after suggested clarifications are addressed

Response: We thank the reviewer for the positive comments and feedback.

Reviewer 2 Report

Rodriguez Bauza and Silveyra have presented a systematic review focused on sex-differences in exercise-induced bronchoconstriction (EIB) in athletes. They found that the overall prevalence of EIB amongst athletes was 23%, which is similar to previous reports, and that the prevalence did not differ on the basis of sex. Additionally, they found that the prevalence of atopy was high in athletes with EIB and that there was a higher prevalence of atopy in male athletes than female athletes. Overall, the study seems well-conducted and the findings are interesting and relevant to the readership of the International Journal of Environmental Research and Public Health. However, I have some concerns relating to the content of the manuscript as well as the analysis and interpretation of the data that should be addressed. Moreover, I have listed specific comments that I hope will be helpful in improving the manuscript.

General Comments:

  1. The introduction includes a great deal of information that isn’t necessarily required to understand the study (g., the definition of minute ventilation, etc.). The authors should consider shortening the introduction to improve readability. This would then allow the authors to expand on more relevant information. Specifically, there should be a dedicated section on sex-differences that would affect the likelihood of EIB in adults (see Becklake MR and Kauffmann F, Thorax, 1999).
  2. As currently written, the purpose of the study is unclear. If the aim is to evaluate the overall prevalence of EIB in men and women, this should be stated clearly.
  3. The prevalence data seem reasonable, but it is unclear why the overall prevalence of EIB was 23% when it was 17% and 13% for males and females respectively. How do the authors reconcile this?
  4. The protocols for determining the presence and magnitude of EIB vary between studies. Can the authors comment on how this may affect the overall findings of the review?

Specific Comments:

Page 1, Line 41: Sex and gender are not synonymous. Sex is certainly relevant in the context of asthma; however, gender is not. Remove “gender” and elsewhere in the manuscript (i.e., Line 68).

Page 2, Lines 68-69: The authors state that “sex and gender differences in the response to exercise have not been established”. This is simply incorrect as there is a significant body of work that has been conducted on the topic. Indeed, there are well documented differences in the cardiovascular, respiratory, thermoregulation, metabolic, and neuromuscular responses to exercise. Please correct this statement accordingly.

Page 2, Lines 72-73: I realize that there are few, but it would be worth discussing these studies in the introduction to provide a better rationale for the study.

Page 3, Lines 120-121: 15% reduction in FEV1 relative to what? Also, the timing and protocol for the detection of EIB is relevant. More information is required, particularly given that this is the primary outcome variable of the review.

Figures: For each Figure, there are abbreviations that should be defined in the legend (e.g., EVH, RCT, CT, etc.). Also replace the word “gender” with “sex”.

Author Response

Reviewer 2

Rodriguez Bauza and Silveyra have presented a systematic review focused on sex-differences in exercise-induced bronchoconstriction (EIB) in athletes. They found that the overall prevalence of EIB amongst athletes was 23%, which is similar to previous reports, and that the prevalence did not differ on the basis of sex. Additionally, they found that the prevalence of atopy was high in athletes with EIB and that there was a higher prevalence of atopy in male athletes than female athletes. Overall, the study seems well-conducted and the findings are interesting and relevant to the readership of the International Journal of Environmental Research and Public Health. However, I have some concerns relating to the content of the manuscript as well as the analysis and interpretation of the data that should be addressed. Moreover, I have listed specific comments that I hope will be helpful in improving the manuscript.

General Comments:

1. The introduction includes a great deal of information that isn’t necessarily required to understand the study (g., the definition of minute ventilation, etc.). The authors should consider shortening the introduction to improve readability. This would then allow the authors to expand on more relevant information. Specifically, there should be a dedicated section on sex-differences that would affect the likelihood of EIB in adults (see Becklake MR and Kauffmann F, Thorax, 1999).

Response: We thank the reviewer for their comments and suggestions. We have modified the introduction to incorporate more information on sex differences (page 2, lines 62-77) and removed the minute ventilation definition.

2. As currently written, the purpose of the study is unclear. If the aim is to evaluate the overall prevalence of EIB in men and women, this should be stated clearly.

Response: We thank the reviewer for their suggestion. We have modified the abstract and last sentence of the introduction to clarify the study’s goal.

3. The prevalence data seem reasonable, but it is unclear why the overall prevalence of EIB was 23% when it was 17% and 13% for males and females respectively. How do the authors reconcile this?

Response: As stated in the manuscript, the majority of studies (including those reporting EIB prevalence) either do not include both sexes or analyze sex differences in EIB prevalence and other outcomes. There could be many reasons for the discrepancy in these numbers, such as the method used to report EIB, which can be problematic or inaccurate, and vary with the conditions in which it is performed, or even present a sex bias in its ability to diagnose EIB. We have discussed this in page 11 lines 284-295. 

4. The protocols for determining the presence and magnitude of EIB vary between studies. Can the authors comment on how this may affect the overall findings of the review?

Response: We thank the reviewer for their suggestion. We have included a sentence addressing this concern in the revised discussion (page 11, line 295-299)

Specific Comments:

Page 1, Line 41: Sex and gender are not synonymous. Sex is certainly relevant in the context of asthma; however, gender is not. Remove “gender” and elsewhere in the manuscript (i.e., Line 68).

Response: We thank the reviewer for their suggestion. We have edited the manuscript accordingly and incorporated definitions of sex and gender and how these relate to the outcomes analyzed in this study (page 2, lines 62-70).

Page 2, Lines 68-69: The authors state that “sex and gender differences in the response to exercise have not been established”. This is simply incorrect as there is a significant body of work that has been conducted on the topic. Indeed, there are well documented differences in the cardiovascular, respiratory, thermoregulation, metabolic, and neuromuscular responses to exercise. Please correct this statement accordingly.

Response: We have edited this sentence to clarify its meaning (page 2, lines 72-77)

Page 2, Lines 72-73: I realize that there are few, but it would be worth discussing these studies in the introduction to provide a better rationale for the study.

Response: We have expanded this section to discuss prior studies (page 2, lines 80-85)

Page 3, Lines 120-121: 15% reduction in FEV1 relative to what? Also, the timing and protocol for the detection of EIB is relevant. More information is required, particularly given that this is the primary outcome variable of the review.

Response: We have edited this section for clarification (page 3, lines 132-135)

Figures: For each Figure, there are abbreviations that should be defined in the legend (e.g., EVH, RCT, CT, etc.). Also replace the word “gender” with “sex”.

Response: We have edited the figure legends accordingly.

Reviewer 3 Report

First, I would like to congratulate the authors of the study. The subject is re-relevant mainly for professionals who work with physical exercises, however many are unaware of the existence of exercise-induced bronchoconstriction, especially in the absence of an asthma diagnosis. Regarding the writing of the manuscript, I considered it clear and objective. However, the discussion seems to me that it could be a little more explored in all points, in order to enrich the study. The limitations of this review are also not clear enough, it would be interesting to add an explanatory topic on this aspect.

Author Response

Reviewer 3

First, I would like to congratulate the authors of the study. The subject is re-relevant mainly for professionals who work with physical exercises, however many are unaware of the existence of exercise-induced bronchoconstriction, especially in the absence of an asthma diagnosis. Regarding the writing of the manuscript, I considered it clear and objective. However, the discussion seems to me that it could be a little more explored in all points, in order to enrich the study. The limitations of this review are also not clear enough, it would be interesting to add an explanatory topic on this aspect.

Response: We thank the reviewer for their positive comments about our manuscript. Following the reviewer’s recommendation, we have expanded the limitations section of the discussion (pages 11, lines 294-305).

Round 2

Reviewer 2 Report

Thank you for addressing my comments. The manuscript is much improved.

Author Response

we thank the reviewer for his/her comments and suggestions